## PERSPECTIVE

# Fermentation technology as a driver of human brain expansion

Katherine L. Bryant [1✉], Christi Hansen[2] & Erin E. Hecht [3✉]

Brain tissue is metabolically expensive. Consequently, the evolution of humans' large brains must have occurred via concomitant shifts in energy expenditure and intake. Proposed mechanisms include dietary shifts such as cooking. Importantly, though, any new food source must have been exploitable by hominids with brains a third the size of modern humans'. Here, we propose the initial metabolic trigger of hominid brain expansion was the consumption of externally fermented foods. We define "external fermentation" as occurring outside the body, as opposed to the internal fermentation in the gut. External fermentation could increase the bioavailability of macro- and micronutrients while reducing digestive energy expenditure and is supported by the relative reduction of the human colon. We discuss the explanatory power of our hypothesis and survey external fermentation practices across human cultures to demonstrate its viability across a range of environments and food sources. We close with suggestions for empirical tests.

**Current hypotheses on metabolic and dietary factors in human brain expansion**. Over the course of 2 million years of evolution, the human brain has tripled in volume. Australopiths possessed brain volumes that were roughly the size of our closest living ape relatives, chimpanzees and bonobos (*Pan troglodytes* and *Pan paniscus*)[1–3]. With the appearance of *Homo*, brain expansion in the human lineage began to accelerate, and continued through to the emergence of *H. sapiens* and *H. neanderthalensis*. Although we have much information on the timeline and extent to which the human brain has expanded in our evolution, the mechanisms which drove this expansion are more difficult to determine. Several theories have been proposed, summarized below.

The Expensive Tissue Hypothesis[4] argues that the expansion of brain size in the hominin lineage required the reallocation of resources from the digestive system. In this view, the limiting factor for brain expansion is the availability of caloric resources, because brain tissue is metabolically expensive compared to most other tissue. Mutations leading to increased brain size, though they might support more adaptive behavior by the organism, would not be adaptive if they carried with them an increased risk of starvation. A reduction in the amount of gut tissue, which has metabolic needs similar to brain tissue, would free up the calories that would otherwise be used to support and maintain digestion and permit its reallocation to the brain. Supporting this model is the fact that in addition to having relatively large brains, the size of the human gastrointestinal tract is 60% of that expected for a primate of our size[4].

However, because gut tissue is itself responsible for extracting nutrients from food, mutations leading to reduced gut size could not be adaptive without a prior shift to a more energy-dense, easy-to-digest food source. Empirical research has supported this model[5–7]. However, some studies across mammalian taxa suggest a more complex relationship with other metabolic investments[8–11]. At the same time, though, when focusing on primates, Isler and van Schaik[12]

[1]Laboratoire de Psychologie Cognitive, Aix-Marseille Université, Marseille, France. [2]Hungry Heart Farm and Dietary Consulting, Conley, GA, USA. [3]Department of Human Evolutionary Biology, Harvard University, Cambridge, MA, USA. ✉email: katherine.bryant@univ-amu.fr; erin_hecht@fas.harvard.edu

found cognitive benefits of a larger brain only increase net fitness if the corresponding energetic costs are accounted for and propose dietary changes as a chief mechanism.

One such proposed dietary change is increased meat eating, which has been argued to have been central to human evolution[13,14]. Analysis of gut morphology in humans suggests it is adapted to both frugivory and carnivory[15]. While modern human diets frequently involve more meat consumption than our anthropoid relatives, and the archeological record shows fossil evidence of butchery in early *Homo*[16,17], some authors[18] argue that a shift to hunting appears later in human evolution—in the Middle to Late Paleolithic. Another possibility is that meat was acquired by other means.

Scavenging after carnivores have finished with a carcass, rather than hunting, may have been the source of meat for human ancestors[19]. Archeological evidence has favored scavenging over hunting[20,21] but evidence from modern hunter-gatherers suggests scavenging is minimally important[22], and analyses of the archeological record indicates that scavenging by early hominins offered low meat yields[23,24]. Bunn and colleagues have proposed that "power scavenging" better explains the patterns of butchery found in the hominin archeological record[25,26]. In this model, human ancestors (*Homo*) confronted carnivores to drive them from fresh kills to obtain valuable portions of meat unavailable to passive scavengers.

Another candidate modification to early hominin diets is the consumption of underground storage organs, or tubers[27]. The importance of foraging in human evolution, linked to the Grandmother Hypothesis, has been highlighted in the tuber-based model of increased calories[28]. The importance of tubers as a source of calories for hominins has been debated, however. One frequently cited source of nutritional data[29] calculated the caloric value of the //ekwa tuber using samples of tubers to determine calories per gram and then multiplying by the total mass of the unearthed tuber. But in the field, Hadza hunter-gatherers discard large fibrous portions of foraged wild tubers prior to consumption[30]. Not only are they labor-intensive to unearth, wild foraged tubers have as little as ¼ of the caloric density reported by Vincent[31], even after cooking.

Another possibility is that the modifications to food through cooking provided the necessary additional calories and nutrients to support a reduction of gut and increase in encephalization[32]. The hypothesis has been extended to encompass others. For example, cooked tubers have been proposed as an important component of the "cooked foods" diet[27,28,32] and it has been suggested that scavenged carcasses were cooked to mitigate microbiological contamination[33]. The trend of reduction of molar size in hominin evolution, perhaps an adaptation from moving from tougher to softer foods[34], fits well with this hypothesis[35].

The benefits of cooking—increase in bioavailability of calories, easier mechanical digestion (especially chewing), and the lowering of energy requirements for digestion—are undoubtable[36,37]. However, there is a lack of archeological evidence for the usage of fire by australopiths and early hominins; the earliest date for the evidence of fire by hominins is frequently cited at 1.5 mya by *H. erectus* during the Middle Pleistocene[38]. Evidence for fire mastery in the Lower Pleistocene still puts this behavior well after the initial emergence of *H. erectus*[39], which is well after selection for brain expansion put hominins on a different course than the *Pan* lineage. While it is likely that the actual origins of human-controlled fire predate its oldest surviving archeological evidence, and older evidence may be newly discovered in the future, mastery of fire technology requires individuals to have the cognitive capacity to plan, create, maintain, and use fire effectively: a tall order for an organism with a brain-to-body ratio barely exceeding modern nonhuman apes. Thus, we should continue to search for other mechanisms that could have kickstarted our ancestors' initial encephalization.

**Hypothesis: external fermentation**. What dietary strategies were accessible by individuals with brains roughly the size of a chimpanzee's? We outline a hypothesis, the *External Fermentation Hypothesis* (Fig. 1). Central to this hypothesis is the realization that the gut *is itself a machine for internal fermentation*: digestion is accomplished via the endogenous microbiome. Culturally-transmitted food handling practices which promoted the externalization of this functionality to the extra-somatic environment could have offloaded energetic requirements from the body creating the surplus energy budget necessary for brain expansion.

We begin with a mechanistic discussion on how external fermentation provides adaptive benefits: it increases macronutrient absorption; it increases the bioavailability of micronutrients, some of which are essential for brain development and function; it supports internal fermentation by the endogenous microbiome; and it provides additional immune benefits. We then present evidence that external fermentation specifically

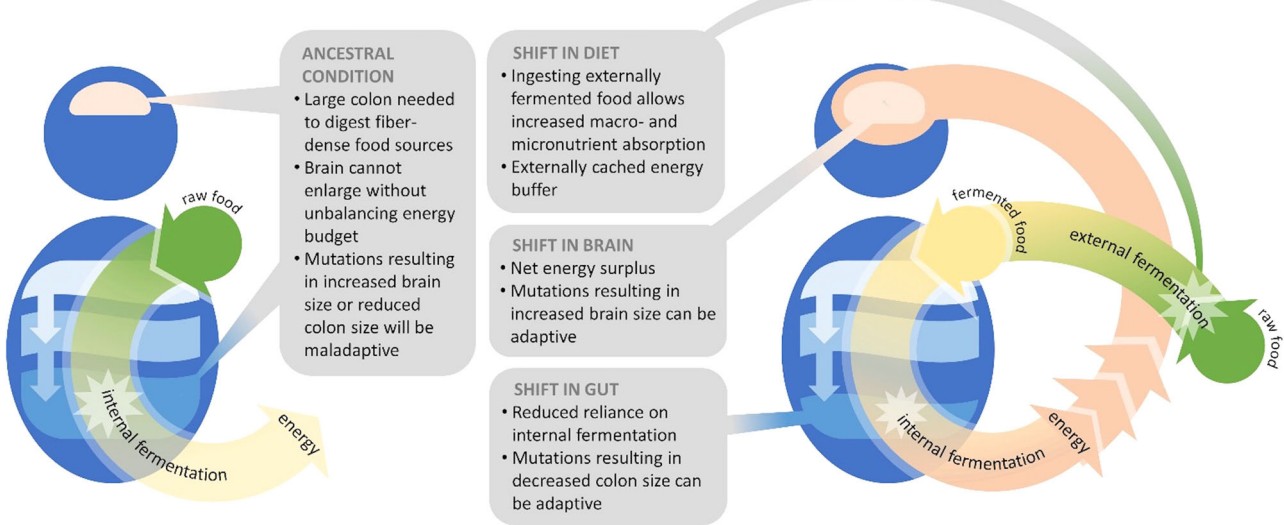

**Fig. 1 The External Fermentation Hypothesis.** A diagrammatic representation of the External Fermentation Hypothesis.

addresses the expensive tissue problem: the reduction in human gut size is attributable mainly to the reduction in the colon, which is the primary site of internal fermentation; furthermore, humans receive a surprisingly low amount of their calories from short-chain fatty acids (SCFAs), which are the products of colon fermentation. Last, we consider the plausibility and explanatory power of the External Fermentation Hypothesis compared to other hypotheses.

**Internal fermentation**. Fermentation is the breakdown of organic compounds by enzymes into alcohol and acids. In the context of human metabolism and nutrition, this enzymatic activity typically originates from bacteria and yeasts. Internal, or gut, fermentation increases the bioavailability of nutrients during digestion.

Digestion is the process of mechanically and enzymatically breaking down organic food matter into macronutrients small enough for absorption through the intestinal barrier and into the bloodstream. The digestion of fibrous, starchy vegetable matter requires a specialized digestive system that supports internal fermentation. In ruminants, this is achieved through additional stomachs; these species are known as foregut fermenters. The hindgut fermenters (humans, other primates, and non-ruminant mammals) evolved a large colon and/or large cecum as a site for internal fermentation and a large surface area for absorption.

In humans, both the large and small intestine contain active, symbiotic bacteria. However, the small intestine contains approximately one million bacteria per mL while the colon contains up to one trillion bacteria per mL[40–42]. Combined with a longer transit time than the small intestine (approximately 1–4 h versus 18–39 h), this means the action within the colon is focused on bacteria-driven fermentation. Although previously it was thought the human colon did little more than resorb water, there is a new focus on the significance of colon for human health, including immune responsivity[43], nutrient absorption, and energy regulation[44].

Internal fermentation increases the body's capacity to absorb macronutrients beyond the normal function of the gastrointestinal tract. Fermented soluble fiber provides an average of 2 cal/g, an additional 50% to the 4 cal/g available from digestible starch and sugars. This energy is only available via the salvaging of otherwise undigested fiber through internal fermentation by gut microbes[45,46]. Fibers are polysaccharide structures that originate primarily in the cell walls of plants; resistant to hydrolyzation by human digestive enzymes, they pass through the small intestine unbroken[47,48]. Once in the colon, these fibers are fermented by enzymes from gut flora, and further degraded by secondary microorganisms into SCFAs[49,50]. Internal fermentation of carbohydrates into SCFAs is estimated to contribute 2-10% of total dietary energy in humans[51–53], but contribute 16% to over 80% of maintenance energy in other mammals (see Table 1).

These internal fermentation products have important biological functions. More than 80% of SCFAs take the form of butyrate, proprionate, or acetate[49]. Butyrate is the preferred energy source for the cells making up the colon wall[47,54,55]; proprionate provides a precursor for hepatic synthesis of glucose and protein[56]; and acetate is used to synthesize cholesterol and other long chain fatty acids, and provides energy to the heart, kidneys, muscle and fat[56–58].

Internal fermentation is critical for the absorption of vitamins and minerals. One way this can occur is via direct synthesis of vitamins by bacteria. In the colon, vitamin K and B-complex vitamins are synthesized by multiple genera of bacteria[58,59]. Lastly, internal fermentation increases micronutrient bioavailability through the breakdown of anti-nutritional factors (ANFs), compounds found in cereals, grains, seeds, legumes, and tubers that bind essential nutrients, preventing their absorption. Phytates and oxalates are chelating ANFs that form complexes with metal cations, preventing the absorption of these minerals[60–63]. Iron, zinc, magnesium, and calcium are particularly impacted by ANFs found in raw plant matter[64], yet sufficient absorption of these is critical for life[65–67].

ANFs are present in the leaves, seeds, and other plant materials that make up a significant portion of many primate species' diets, including hominoids. Foraging strategies of primates suggest deliberate avoidance of plant species with high endogenous ANF content, as well as preference for younger leaves to reduce ANF burden and increase digestibility[68,69]. Primates that have folivory-heavy diets have evolved gut specializations for internal fermentation—either through the evolution of a complex forest-omach, as in colobine monkeys[70] or through the expansion of the hindgut (cecum and colon)[71]. Predictably, hindgut fermenters have cecum/colon volumes that correlate positively with the proportion of leaves that make up their total diet[72]. We propose that external fermentation may represent a parallel, alternative adaptation.

**External fermentation**. Rather than relying on the microorganisms within the gut, external fermentation is carried out by organisms in the environment or on the surface of the organic material itself. Like internal fermentation, external fermentation increases the bioavailability of ingested nutrients, specifically, the absorption of macronutrients and micronutrients. In addition, external fermentation contributes to the health and efficacy of the host's gut microbiome, in turn, facilitating nutrient absorption.

**Table 1 Energy derived from short-chained fatty acids produced by gut fermentation.**

| Species | Latin name | Diet | Total % of energy | Citation |
|---|---|---|---|---|
| Cattle | *Bos taurus* | Ruminant herbivory | 72% | 131 |
| Sheep | *Ovis aries* | Ruminant herbivory | 84% | 132–134 |
| Pony | *Equus ferus caballus* | Monogastric herbivory | 30% | 135 |
| Rabbit | *Oryctolagus cuniculus* | Monogastric herbivory | 32% | 136–138 |
| Beaver | *Castor canadensis* | Monogastric herbivory | 19% | 139 |
| Porcupine | *Hystrix dorsata* | Monogastric herbivory | 16% | 140 |
| Pig | *Sus scrofa* | Omnivory | 36% | 141–143 |
| Rat | *Rattus norvegicus* | Omnivory | 5% | 144 |
| Mantled howler monkey | *Alouatta palliata* | Monogastric herbivory | 30% | 145 |
| Gorilla | *Gorilla gorilla gorilla* | Monogastric herbivory | 57%* | 45 |
| Human | *Homo sapiens sapiens* | Omnivory | 2–10% | 51–53 |

Percentage of maintenance energy derived from the production of short-chained fatty acids (SCFAs) via gut fermentation. Information adapted chiefly from Bergman[146].
*Values for gorillas were estimated from diet composition and human colonic fermentation rates.

External fermentation enhances digestibility of carbohydrates and proteins. Fermentation of legumes hydrolyzes macromolecules into more easily digestible individual amino acids[73] and sugars[74]. These benefits have led public health scholars to recommend increasing the consumption of fermented foods in countries experiencing food insecurity and high infant mortality[75,76].

External fermentation also improves the bioavailability of micronutrients in a number of ways. B-complex vitamins produced from the external fermentation of carbohydrates can increase the amounts of B vitamins (thiamin, riboflavin, and niacin) by up to 10-fold[77,78]. External fermentation can also break down ANFs.

Phytate, a chelating ANF, can be broken down by phytase, an enzyme that some mammals—but not humans—have evolved the ability to produce endogenously[79]. External Lactobacillus-driven fermentation is an alternative to phytase: by lowering the pH, it provides a favorable environment for both bacterial and endogenous phytase to hydrolyze bound phytate and release minerals[80]. Oxalate, another chelating ANF, and tannins, ANFs which bind to and lower the bioavailability of proteins, can also be degraded through external Lactobacillus fermentations[81,82]. Of note, phytate is more effectively degraded by external fermentation than by cooking, as phytase bioactivity decreases above 80 °C[83,84].

External fermentation can go further than simply increasing nutrient bioavailability. It can also render poisonous foods edible. One example is the detoxification of cyanogenic glycoside in bitter cassava (also known as yuca or manioc), a common staple for hundreds of millions of people living within the Tropical Belt[75,85]. If consumed unfermented, cassava's cyanogenic glycosides are hydrolyzed by colonic microorganisms and absorbed as cyanide, causing convulsions, hypotension, respiratory failure, decreased heart rate, and death[85,86]. When processed properly, cell walls in the cassava tuber are broken down by Lactobacillus bacteria, permitting endogenous enzymes normally sequestered from the cyanogenic glycosides to hydrolyze the toxin. The production of lactic acid during fermentation also acidifies the environment and provides a favorable milieu for other microorganisms to contribute to the hydrolysis of up to 95% of the toxin prior to consumption[85,87].

The third mechanism by which external fermentation supports digestion is by supporting and contributing to the gut microflora, which in turn contributes to ongoing enhanced nutrient absorption. Ingested microflora from fermented food colonize their new environment, contributing diversity to the host microflora and boosting the guts' ability to ferment more polysaccharides into energy and nutrients[54,56], although the extent of incorporation of microflora in the gut is dependent on multiple factors[88]. Ingested probiotic bacteria also support the health of endogenous microflora by producing bacteriocins, toxins that competitively inhibit pathogens[89,90]. Even transient contact with certain species of microorganisms is enough to beneficially alter existing colonies of bacteria or produce anti-pathogenic metabolites[90]. This may effectively act as an external reservoir of bacteria necessary for internal fermentation. In many primate species, this reservoir function is supplied internally by the cecum[91,92]. Cecal size is larger in Old and New World monkeys and prosimians than in anthropoids, smaller in cercopithecoid monkeys, and reduced further in hominoids; of the great apes, humans have the most reduced cecum[93].

By supporting the gut flora responsible for internal fermentation, external fermentation may also help protect the host from infection and disease. Once bound to colonic epithelial cells, probiotic bacteria impede pathogenic bacteria from colonizing the intestinal wall, reducing their ability to penetrate the bloodstream[54,90]. A healthy colon microbiome producing large amounts of SCFA through the fermentation of indigestible carbohydrates is well-linked to decreased inflammation in the gut and a reduction in gastrointestinal disorders[94]. As colonic epithelial cells derive the majority of their energy from SCFAs, diets low in plant fiber force colonic microorganisms to rely on dietary fats and protein, resulting in decreased SCFA production. In the absence of adequate fiber, microbes may degrade the epithelial mucus layer which can lead to sepsis[95].

To summarize, then, the ingestion of externally fermented foods provides four critical components to digestion and absorption. First, it increases the digestibility of foods; second, it increases the bioavailability of micronutrients; third, it supports gut fermentation by contributing to host microfloral diversity; and lastly, it supports immune function and resistance to disruption of the gut microbiome. These benefits would have been adaptive advantages for our early ancestors and could have played a key role in human brain evolution, as we describe below.

**External fermentation as a driver of early hominin brain expansion.** The development of external fermentation technology represents a plausible metabolic mechanism leading to brain expansion beginning at our ancestors' divergence from the australopiths. The Expensive Tissue Hypothesis posits that the reduction of gut tissue in the human lineage permits the reallocation of metabolic resources towards brain tissue, which is metabolically expensive[4]. The obvious paradox here is that gut tissue, while metabolically expensive as well, is the site of caloric uptake for the organism. Thus, reduced gut sizes could only evolve if our ancestors were able to exploit a more nutrient-dense and easily digestible food source. Aiello and Wheeler examined the relative proportion of the most metabolically expensive tissues outside of the brain: the heart, liver, kidneys, and gastrointestinal tract, and found the gastrointestinal tract was 60% smaller than predicted for a primate of our size[4]. Taking a closer look at the gastrointestinal tract, we observe the reduction in size is not equal across organs. Colon volume in non-human great apes is twice that of the small intestine (in gorillas, close to five times the volume); whereas in humans, the ratio is reversed, with the colon having approximately one-third the volume of the small intestine[14,96].

Using estimations from Milton[96–98] on differences between the proportions of small intestine and colon in humans and apes, we calculated the approximate masses of these subcomponents by taking the midpoint values given by Milton[14] and applying them to the total gastrointestinal tract values from Aiello and Wheeler[4]. Table 2 shows these calculations; Fig. 2 shows the relationship between organ sizes in a hypothetical 65 kg human with ape-like organ sizes (expected) and the actual proportions in modern western humans (actual). While total gut reduction is impressive (a reduction of over 41%), the reduction is not consistent across subcomponents. Small intestine proportion actually increases, from approximately. 4 kg to. 62 kg in modern humans, an increase of 58%. The subcomponent which accounts for the largest share of the reduction is the colon. With a predicted ape-like value of 0.85 kg, a typical human instead has an estimated mass of. 22 kg, a reduction of 74%—the largest reduction of any of the gut subcomponents and any of the other major organs (Table 2).

A smaller colon may reflect a reduction of dependence on fibrous plant material, given that a major function of the colon is to house bacteria that aid in the breakdown of enzyme-resistant carbohydrates to SCFAs. Did a shift to meat-eating, as suggested by Milton, permit this drastic reduction in colon size in the human lineage? Indeed, humans and members of the order

| Table 2 Expected masses of major organs in humans based on great ape values and comparison with observed values. | | | | | | |
| --- | --- | --- | --- | --- | --- | --- |
| | **Heart** | **Kidney** | **Liver** | **Other Gut** | **Small Intestine** | **Colon** | **Brain** |
| Expected | 0.32 | 0.238 | 1.563 | 0.63 | 0.404 | 0.846 | 0.45 |
| Actual | 0.3 | 0.3 | 1.4 | 0.26 | 0.616 | 0.22 | 1.3 |
| % change | −6.3% | 26% | −10.4% | −58.1% | 52.3% | −74.0% | 188.9% |

Expected masses in kg and percentage difference for major organs in a hypothetical 65 kg human based on great ape values (expected) and observed (actual) measurements in western humans. Data based on Aiello and Wheeler's (1995) compilation of data from Stahl[147], Stephan et al.[148], and Chivers and Hadlick[72]. Gastrointestinal tract weights were subdivided based on ratios from Milton[97,98,149].

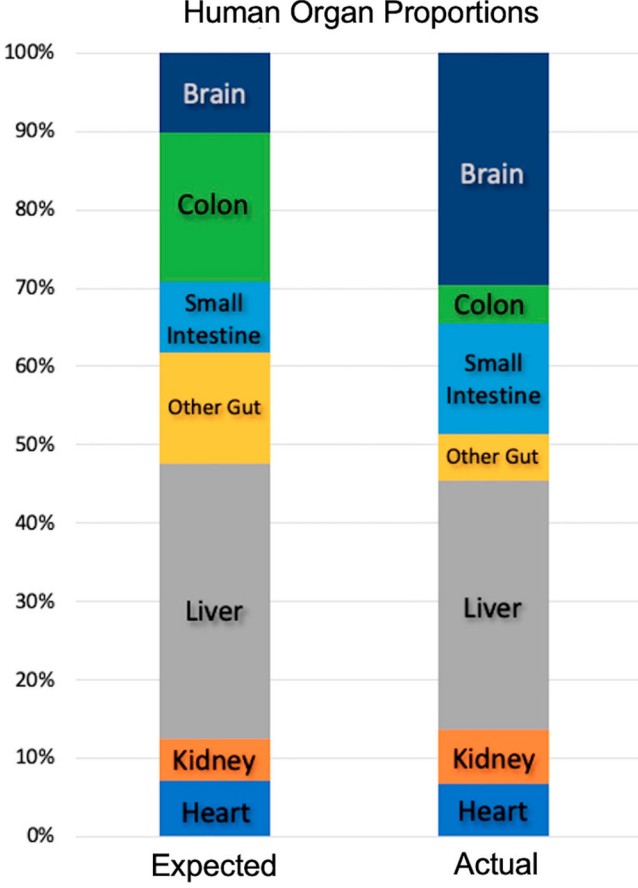

**Fig. 2 Estimated proportions of major organs in a modern Western human.** Proportions of major organs in a hypothetical 65 kg modern Western human using data from Table 1. "Expected" represents the ratio of organ masses expected if humans had proportions in line with other great apes. "Actual" represents an estimation of the ratios in a typical modern Western human.

Carnivora share a small colon size. However, the gut transit time in Carnivora is much faster than in humans. Although Milton postulates that this difference is due to our evolutionary history as plant eaters[96], another explanation is that colon reduction follows from a reduced need to break down fibrous plant material within the digestive tract due increased bioavailability of nutrients *before* food is consumed—i.e., external fermentation (Fig. 1).

Is external fermentation a realistically plausible strategy for our australopith ancestors? A major hurdle is that it requires a cache of food to be stored in a location conducive to fermentation and remain there for a sufficient duration. The transport and caching of food is something that separates human ancestors from our closest extant primate relatives. Early hominins appeared to have carried food resources and stone tools to specific locations, up to 10 kilometers away[99,100]. Combined with the accumulating

evidence that stone tools were likely knapped prior to the emergence of *Homo*[101], it has been argued that australopiths were knapping tools, butchering animals, and carrying and caching both food and tools[99,102,103]. By contrast, although chimpanzees do occasionally transport tools, distances are frequently less than 500 meters and rarely reach a kilometer[104]. Food transport is limited to the transport of meat across short distances; most other food sources are eaten where they are acquired[105].

Forethought and mechanistic understanding are *not* requirements for the initial emergence of external fermentation. Our early ancestors may have simply carried food back to a common location, left it there, and intermittently eaten some and added more. Re-use of a storage location could have promoted the stability of a microbial ecosystem conducive to fermentation. As new food items were brought back and added to the cache, they could have become inoculated with the microorganisms already present in the location (or on the hominids themselves).

External fermentation may have occurred for a protracted period of time in this manner—as an epiphenomenon of pre-existing adaptive habits of food transport and storage. Socially-transmitted practices such as the re-use of the same storage locations, containers, or food-processing tools would have further promoted the initiation of fermentation and the stability of ongoing ferments. Over time, additional facilitation may have come from culturally reinforced norms, such as superstitions about where food must be stored or how long it must rest before being eaten. As brain size and cognitive capacity increased, understanding of the proximate causes and consequences of fermentation could have progressed in a gradual fashion. Strategic control of fermentation practices would have become increasingly complex, up to the modern day, where cumulative culture has produced a remarkable diversity of fermentation practices (see supplementary Table 1).

**Explanatory power compared to other hypotheses**. The emergence of meat-eating, tuber-harvesting, and cooking have all been proposed to account for human brain expansion; why should our just-so story be given any additional credence? Below, we consider several explanatory advantages of the External Fermentation Hypothesis versus other current hypotheses.

*Less brain power required*. In searching for an initial trigger to the upward spiral of human brain expansion, it is important to recognize that it would have to occur in organisms with brains roughly the size of a chimpanzee. The cognitive capacities of chimpanzees may arguably be less complex than those of australopiths, particularly later, larger-brained australopiths. At a minimum, we can reason that behaviors which are well within the chimpanzee repertoire were likely attainable by australopiths, and behaviors beyond this repertoire may have at least been challenging for australopiths. The possibility that ingestion of externally fermented foods has deep roots has been proposed by others. Carrigan and colleagues[106] suggest that the Pan-Homo last common ancestor targeted a broad range of spontaneously fermented fruits on the forest floor, while Amato and co-

workers[107] propose a more specific focus on fruits with chemical and/or physical defenses that would otherwise make ingestion difficult or problematic. Amato and colleagues further suggest that Ardipithicus and early *Homo* both additionally incorporated fermented tubers[107]. Dunn and colleagues[108] hypothesize that *H. erectus* may have been engaged in food fermentation.

Chimpanzees display a variety of complex, socially learned, instrumental behaviors oriented toward food, such as "fishing" for termites or honey using sticks, and fashioning spears to hunt monkeys. A well-studied example is chimpanzee nut cracking. Juvenile chimpanzees spend years learning to accomplish this using a hammer stone and anvil stone. During this time, they make errors like banging the hammer stone on the anvil stone while the nut is left resting on the ground nearby[109]. This suggests that chimpanzees have difficulty understanding the underlying causal mechanism—i.e., that the nut's shell is opened *because* it was struck. Despite nut-cracking occurring in a social context with multiple expert and novice crackers in the same location, using the same tools, at the same time, understanding of the *causal relationship* between percussion and a cracked shell is not socially learned. Instead, each chimpanzee independently "re-discovers" this causal relationship for itself. The social context merely contributes a scaffold in which independent learning can occur[110].

Chimpanzee stone tool use has continued substantially unchanged for at least 4,300 years[111]. Thus, animals with brains similarly sized to australopiths are capable of socially transmitting instrumental behaviors which are stable over long periods of time in the absence of underlying causal understanding about how the specific details of the action are related to its end goal. Aspects of behavior that *are* easily socially transferred by chimpanzees include memory for the objects, tools, and locations that are involved in achieving a particular goal. We propose that this is all that is required for social transmission of fermentation to take hold.

In comparison with fermentation, the means-ends dependencies between objects, actions, and outcomes in cooking are considerably more constrained and complex. Cooking requires comprehension of causal mechanisms between multiple interacting objects—i.e., a chain of sequential, dependent interactions between fuel, flames, and raw food. This is precisely the type of means-ends dependency that is challenging for chimpanzees. Thus, we propose that external fermentation poses less of a cognitive hurdle than control of fire and is thus more likely than cooking to have impacted the gut-brain tradeoff at an earlier point in evolution.

Notably, one experiment did address whether chimpanzees might have some of the cognitive skills necessary for cooking. When chimpanzees were presented with a device which, via unseen experimenter manipulation, "transformed" raw food to cooked food, chimpanzees deliberately used the device to obtain the latter[112]. Beran and colleagues[113] argue that this experiment reveals more about chimpanzees' food preferences and capacity for bartering or exchange behavior than it does about their capacity for cooking. We propose that these results provide evidence that chimp-sized brains are capable of understanding and performing the steps required to ferment food: put food in a particular place, wait for it to become transformed, and then enjoy an improved version.

*No lightbulb moment required.* While the utility of fire and fermentation for food processing could both be discovered accidentally, a spontaneous discovery was more probable for fermentation. Naturally occurring fire is not a daily incident, and opportunities for our ancestors to spontaneously notice its potential for cooking must have been sporadic. Although accidental cooking may have occurred (for example, the action of

wildfire on animal carcasses or buried tubers), the transition from opportunistic, infrequent access to accidentally-cooked food to a long-term and stable source of extra calories would require a "lightbulb moment:" recognition of the effects of the accidental process, and intentional, deliberate actions to reproduce their causes. In contrast, naturally occurring fermentation *is* a daily incident. Bacteria and fungi are everywhere, all the time, and spontaneously colonize food; no "lightbulb moment" is required to transform unintentional external fermentation into a source of extra calories.

*Environmental stability.* Fires require ongoing active effort to maintain, whereas fermentation is largely a passive process. Once started, an ongoing fermentation does not extinguish, and does not require tending or restarting, as fire does. Moreover, this environmental persistence offers more chances for social learning, potentially further supporting the longevity of the practice across generations.

*Stable food preservation—a caloric buffer.* Because brain tissue is so energetically expensive, and is intolerant of reduced energy availability, organisms with larger brains are more susceptible to fluctuating availability of food[8]. The evolution of increased adipose tissue in humans is a proposed adaptation to ameliorate this risk, as fat provides an "internal buffer" for survival through lean times[11,114]. External fermentation practices may have provided a secondary, "external buffer." Fermentation can preserve food for *years*. Food spoilage is caused by microorganisms, and some of the best inhibitors of microorganisms are other microorganisms. Fermentation allows for the proliferation of non-harmful or beneficial strains which out-compete harmful strains.For example, by-products of fermentation include alcohol and acid, which inhibit further microbial growth, effectively preserving the food. There are other food storage techniques whose effective time-scales are within that of fermentation, such as smoking, drying, freezing, and salting (notably, often used in combination with fermentation). However, compared to these other methods, we propose that fermentation may have been accomplishable more easily, across a wider range of environments, and by earlier, smaller-brained, less cognitively-complex ancestors.

*Summary of explanatory power of the External Fermentation Hypothesis.* Unlike other proposed dietary modifications, a transition to eating fermented foods does not require great leaps in cognitive ability. It does not require advanced planning, as hunting, particularly hunting in groups, would. It does not require the acquisition of a difficult technology, as in fire for cooking. It can more directly explain, than tubers, meat, or cooking, how colon fermentation could be replaced through dietary changes.

Fermentation accounts for all the benefits that cooked food offers: softer food, higher caloric content, greater bioavailability of nutrients, and protection from pathogenic microorganisms. Fermentation solves several problems. It does not require special materials beyond a place to store food (a hollow, a cave, or a hole in the ground could work). It does not require overcoming fear—there is a low barrier to entry. It can be stumbled upon rather than requiring planning and tool use. And it does not require, initially, long-term planning, focused attention, or sophisticated social coordination.

In all likelihood, for most of human history, it was nearly impossible to store food for any length of time *without* bacterial or fungal growth. Life-threatening illness is a risk of some food-borne microbes (e.g., *E. coli*, *salmonella*). Thus, it would have been necessary to either keep *all* microbial growth below potentially toxic levels (via e.g., drying, salting, smoking, or

freezing), or encourage high levels of "good" microbial activity to out-compete the bad. The latter seems clearly easier.

**Contemporary human fermentation practices**. Current fermentation practices can provide insight into its role in our past. We have created a detailed list of examples that provide a sense of the widespread scope and impact of fermentation technology on the human diet worldwide (Supplementary Table 1). Humans deliberately ferment foods of nearly every kind, including fruits, vegetables, grains, legumes, animals (muscle meat, organs, fat and bones), dairy, fish, and shellfish. Fermentation is practiced successfully in a diversity of climatic contexts, from tropical humid conditions to arctic environments. It is accomplished with a wide variety of microorganisms, including bacteria, filamentous fungi, and yeasts. Moreover, fermentation works on a range of time-scales from hours to years, effectively acting as a short-term flavor enhancer or a long-term storage technique. Our survey represents what is likely a relatively shallow and sparse representation of the full breadth of modern, historical, and pre-historical fermentation practices. For example, Neanderthals are proposed to have fermented meat to preserve vitamin C and thereby avoid scurvy[115]. This variability of fermentation practices represents a clear opportunity for more probative ethnographic and cultural evolution research both broadly across human populations[116] as well as specific ethnographic analyses into the role that fermented meats play in pre-Industrial cultures in the tropics[117].

We present this aggregation of examples as evidence supporting three points. First, given the incredible range of food types and environments that can lead to successful fermentation, it is plausible that fermentation was also possible for the food types and environments of early human ancestors. Second, it seems that fermentation is ubiquitous across extant cultures and can be considered a human universal. This is consistent with fermentation having a very early emergence. Third, while cultural practices for fermenting food vary across the globe, it seems clear that humans in general have a taste for fermented food. This preference may be an evolved mechanism which emerged because an attraction to these flavors was adaptive in our shared past. Notably, many fermented foods listed in Supplementary Table 1 such as fish sauce, soy sauce, and vinegar, are *condiments*—i.e., substances added to other food items mainly for the purpose of improving palatability.

**Testing the External Fermentation Hypothesis**. If our hypothesis is correct, then we might expect to find evolved innate preferences for beneficial fermentation products or evolved innate aversions to dangerous byproducts of "off" fermentation. Interestingly, it appears that many of the most disparately-regarded foods—seen by some as prized delicacies, and by others as supremely unappetizing—are fermented: for example, thousand-year eggs, natto, and Limburger cheese. These preferences appear to be highly culturally specific, which might be adaptive given the high cultural diversity of fermentation practices and the risks of consuming a ferment gone awry. The same flavors or odors which might signal "good" food in one culture could emanate from "off" ferments in another. Future research could address the extent to which preferences for fermented products are innate, cultural, or may be the product of gene-culture coevolution[118]. For example, sour taste abilities have been proposed to have co-evolved with the production of fermented foods[119]. Notably, preferences for sour or acidic foods are relatively rare in the animal kingdom[119]. Human food preferences are highly variable across individuals and cultures and are culturally learned, a phenomenon which may be adaptive[120]. Are preferences for fermented foods more susceptible to cultural learning than other food preferences? Are

they more sensitive to experience in a developmental critical period, and/or less flexible after this period closes? Are they heritable, either genetically or epigenetically[121]?

Fermented foods have the potential to be colonized by pathogenic microbes. How might the risks and benefits of external fermentation compare to the risks and benefits of other potential solutions to balancing the metabolic budgetary increase associated with brain enlargement? Hunting, scavenging from large carnivores, and fire use carry their own risks; perhaps the risks of fermentation were more predictable and thus more reliably mitigable through individual and cultural learning. In the environments and time periods relevant for our hypothesis, what situations might have caused a fermentation to become pathogenic? How easy would it have been for a hominid with a chimpanzee-sized brain to avoid these risks, either deliberately or via socially-learned practices? How often would "off" fermentation have catastrophic results versus temporary illness, and how would this have compared to injuries sustained during hunting, scavenging, or fire use? Potential answers to these questions might come from food microbiology investigations where fermentation products are studied under varying environmental conditions, or from field research with existing hunter-gatherer populations. At the same time, hosting large microbial communities within the colon likely carries its own risks, including increased risk of colonization by pathogenic microbes and increased host metabolic costs associated with immune monitoring of these communities. Reducing internal microbial load might attenuate these risks and costs, but empirical research is necessary to directly probe this cost-to-benefit ratio.

Examinations of the human microbiome could provide evidence for or against the External Fermentation Hypothesis. A comparative analysis with chimpanzees, bonobos, and gorillas found the human microbiome has undergone accelerated deviation from the ancestral ape state, and now shows reduced diversity[122], which is consistent with modern increased reliance on commercially produced food but may also be consistent with earlier increased reliance on external microbial communities. The human microbiome also appears to have undergone alterations associated with our species' increased sociality[108]. If early humans really offloaded internal fermentation to the external environment, we should expect to see changes in the internal microbial community associated with this shift. Would internal species associated with a particular food become less abundant over time, while the external species proliferated? Would humans' internal flora adapt to now specialize in the post-fermentation product, perhaps with evolved adaptations for tolerating higher levels of fermentation by-products like acid or ethanol? Can we trace the co-evolution of gut flora and external fermentation flora as human populations have moved around the globe? Could phylogenetic analyses of human gut microbes provide a window onto the onset of fermentation practices in human evolution? Additionally, our hypothesis predicts that the human colon has a smaller microbial population than that of our ape relatives, which offers a target for empirical testing.

Genetic and genomic analyses focused on the perception of fermented foods offer opportunities for additional empirical tests. One potential target is olfactory receptor (OR) genes. Our hypothesis predicts that the human lineage may have experienced positive selection on OR genes that detect fermentation products. One analysis found 6 functional ORs showing evidence of positive selection in the human but not chimpanzee lineage, and 5 showing the reverse; two of each of these are located in the OR5 family at 11p15.4[123]. Most ORs are "orphans," meaning the natural ligand (odorant) is unknown, but adjacent OR genes tend to detect similar compounds, and ORs at this locus generally detect n-aliphatic odorants[124]. De-orphaned OR5 genes respond to methyl octanoate, which has a fruity odor, is found in fruit wines,

and can be produced by *S. cerevisiae*[125], and methylvaleric acid, which is a key aroma compound in aged cheese[126]. We might also expect relaxed selection in the human lineage with ongoing positive selection in the chimp lineage for toxic or anti-nutritive compounds which are reduced by fermentation (e.g., oxalate, phytates). Interestingly, these compounds are bitter, and the human lineage has experienced relaxed constraint on the *TAS2R* gene family, which encodes for bitter taste receptors[127]. Modern human populations show variation both in TAS2R loci and in the ability to detect bitter taste[128,129]. While this evidence is suggestive, it is indirect. Additional research might leverage more probative analyses of selection in human and extinct hominin genomes or examine whether *OR* or *TAS2R* variation can be linked to preferences for fermented foods.

A further possibility is to examine genetic shifts associated with digestive, metabolic, and immune processes that may be impacted by an increased reliance on external fermentation. Notably, nonhuman apes, who do ingest fermented fruits[107], show alterations to the ADH4 gene linked to ethanol processing[106]. The capacity to metabolize ethanol long predates the onset of hominin brain expansion and may have been associated with the transition from an arboreal to a terrestrial lifestyle as much as 10 mya[106]. For example, one genetic shift shared by humans and other great apes is the emergence of an additional hydroxycarbolic acid receptor, $HCAR_3$, in addition to the two that are present in other primates. $HCAR_3$ is activated by D-phenyllactic acid, which is an antimicrobial compound produced by lactic acid fermentation and present in sauerkraut at sufficient levels to trigger $HCAR_3$ activation and its downstream effects including regulation of immune and energy functions[130]. Because alterations to ADH4 and $HCAR_3$ were likely present in the last common ancestor of all extant hominids, they may represent preconditions for a reliance on external fermentation. Future work could examine whether genetic change has occurred in loci involved in metabolizing compounds in fermented foods ingested by humans but not other apes.

**Conclusions**. We have proposed that the acquisition of fermentation technology by early hominins—the External Fermentation Hypothesis—is a good candidate mechanism for human brain expansion and gut reduction. The offloading of gut fermentation into an external cultural practice may have been an important hominin innovation that laid out the metabolic conditions necessary for selection for brain expansion to take hold. While the potential importance of fermentation in the evolving human diet has recently been postulated[108], and the reduction in human colon size has been previously observed[14], to the best of our knowledge, the possibility that external fermentation served as the initial trigger in the human lineage for the expansion of brains and the reduction of the gut—specifically, the colon—has so far been unnoticed. We have discussed the adaptive benefits of this hypothesized scenario, its realistic plausibility, and its explanatory power relative to other hypotheses. We invite commentary and experimental tests from the broader academic community.

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

## Acknowledgements

The authors would like to thank Matthew C. Maddox for his theoretical contributions early in the development of this hypothesis through his expertise in fermentation technologies. The authors thank Christina Rogers Flattery, Brittany Howell, and Dan Lieberman for their valuable feedback on earlier versions of this manuscript.

## Author contributions

K.L.B. and E.E.H. conceived the paper, K.L.B. and E.E.H. compiled data and analysis, K.L.B., E.E.H., and C.H. wrote the manuscript, with E.E.H. focusing on metabolic and nutrition components, E.E.H. on human evolution components, and K.L.B. on fermentation and culture components. All authors contributed to the final editing process.

## Competing interests

The authors declare no competing interests.
