## [Peer Review File · Communications Biology]

Reviewers' comments:

Reviewer #1 (Remarks to the Author):

This paper argues that human brain evolution was supported by diets that included external food fermentation. It outlines the benefits of fermentation generally (both internal and external) and then examines the extent to which the consumption of externally fermented foods can explain human brain evolution (in contrast to other existing hypotheses). Overall, the arguments make sense. However, I think the presentation of material could be strengthened. Some of the headings used, particularly on the first page, are unusual. If this is not part of the journal format, I would suggest removing some of the headings to make the story flow a little more. I also think a more robust separation and description of internal versus external fermentation and the benefits and drawbacks would make the story stronger. Finally, there are a few papers that I think the authors should consider reading and including/building upon as many of the arguments are related. These changes will add more depth to the arguments and make the contribution stronger overall.

Detailed comments:

Background: I would suggest the authors better separate internal and external fermentation since an important part of the argument seems to be that humans shifted from internal to external. As written the background mixes ideas from both fermentation types, and it gets difficult to separate them. I realize the underlying process is similar in terms of benefits, etc., but I think making the separation early will help make points later.

Discussion: Section A might benefit from some discussion of the potential costs of hosting large microbial communities internally. Not all microbes are beneficial, and even those that are require both resources and immune monitoring from hosts and therefore incite metabolic costs. Some exploration of these potential costs might nicely complement the discussion of intestinal dimensions and metabolic costs. I wonder if there are data describing the estimated microbial biomass in human colons compared to other primates - or could it be extrapolated using colon size...? I suppose this could also go into the Background?

I am not sure if the current headers of Introduction, Background, and Discussion make sense. It may be enough to use some of the subheadings. The hypothesis/thesis heading is unusual, and some of the subheadings in the Introduction seem unnecessary (or could be used instead of the overall Introduction heading).

I would suggest the authors look at the following papers for more ideas/evidence:

<https://www.journals.uchicago.edu/doi/full/10.1086/715238>
<https://onlinelibrary.wiley.com/doi/full/10.1002/ajpa.24257>
<https://www.frontiersin.org/articles/10.3389/fevo.2020.00025/full>
<https://royalsocietypublishing.org/doi/full/10.1098/rspb.2021.1918>
<https://www.pnas.org/doi/abs/10.1073/pnas.1404167111>
<https://journals.plos.org/plosgenetics/article?id=10.1371/journal.pgen.1008145>
<https://www.journals.uchicago.edu/doi/full/10.1086/716014>
<https://paleoanthropology.org/ojs/index.php/paleo/article/view/114>
<https://www.sciencedirect.com/science/article/pii/S1040618218311698>

Reviewer #2 (Remarks to the Author):

The authors present a logical hypothesis regarding an additional source of increased energy availability required to facilitate brain expansion in the human lineage – fermented foods. Their descriptions are thorough, and I appreciate their suggestions for future comparative studies. However, this manuscript is missing a discussion of work that has already focused on the importance of fermented foods in nonhuman ape and human evolution, including: 1) alterations to the ADH4 gene that improve ethanol digestion (from fermented fruits) in apes (Carrigan et al. 2015, PNAS); 2) coevolution between fermented foods and sour taste abilities (Frank et al. 2022, Prob B); and 3) the possibility that Neanderthals fermented meat to maintain vitamin C content and avoid scurvy (Speth 2019, Quarterly Int'l).

Response to Reviewers' comments (our response in bulleted indents)

Reviewer #1 (Remarks to the Author):

This paper argues that human brain evolution was supported by diets that included external food fermentation. It outlines the benefits of fermentation generally (both internal and external) and then examines the extent to which the consumption of externally fermented foods can explain human brain evolution (in contrast to other existing hypotheses). Overall, the arguments make sense. However, I think the presentation of material could be strengthened. Some of the headings used, particularly on the first page, are unusual. If this is not part of the journal format, I would suggest removing some of the headings to make the story flow a little more. I also think a more robust separation and description of internal versus external fermentation and the benefits and drawbacks would make the story stronger. Finally, there are a few papers that I think the authors should consider reading and including/building upon as many of the arguments are related. These changes will add more depth to the arguments and make the contribution stronger overall.

Detailed comments:

Background: I would suggest the authors better separate internal and external fermentation since an important part of the argument seems to be that humans shifted from internal to external. As written the background mixes ideas from both fermentation types, and it gets difficult to separate them. I realize the underlying process is similar in terms of benefits, etc., but I think making the separation early will help make points later.

- Thank you for taking the time to review our article. Following this suggestion, we have adjusted the introduction to more clearly separate coverage of internal and external fermentation (see Introduction, Section III: Internal Fermentation and Section IV: External Fermentation; pages 2-4).

Discussion: Section A might benefit from some discussion of the potential costs of hosting large microbial communities internally. Not all microbes are beneficial, and even those that are require both resources and immune monitoring from hosts and therefore incite metabolic costs. Some exploration of these potential costs might nicely complement the discussion of intestinal dimensions and metabolic costs.

- Thank you for this suggestion. We have added these points to the discussion, specifically, Section IV: Testing the External Fermentation Hypothesis (end of paragraph 2 and beginning of paragraph 3; and a new paragraph 5 which discusses the putative relationships between these metabolic costs and genetic changes in the human and hominid lineage that may have facilitated the move from internal to external fermentation).

I wonder if there are data describing the estimated microbial biomass in human colons compared to other primates - or could it be extrapolated using colon size...? I suppose this could also go into the Background?

- We do not know of any current data describing the microbial biomass in human colons versus other primates, but this is a good empirical test of our hypothesis.

I am not sure if the current headers of Introduction, Background, and Discussion make sense. It may be enough to use some of the subheadings. The hypothesis/thesis heading is unusual, and some of the subheadings in the Introduction seem unnecessary (or could be used instead of the overall Introduction heading).

- We have removed these subheadings and renamed the headers. The new organization is: I. Current Hypotheses on Metabolic and Dietary Factors in Human Brain Expansion; II. A New Hypothesis: External Fermentation; III. Internal Fermentation; IV. External Fermentation; V. External Fermentation as a Driver of Early Hominin Brain Expansion; VI. Explanatory Power Compared to other Hypotheses; VII. Contemporary Human Fermentation Practices; VIII. Testing the External Fermentation Hypothesis; IX. Conclusions.

I would suggest the authors look at the following papers for more ideas/evidence:

<https://www.journals.uchicago.edu/doi/full/10.1086/715238>
<https://onlinelibrary.wiley.com/doi/full/10.1002/ajpa.24257>
<https://www.frontiersin.org/articles/10.3389/fevo.2020.00025/full>
<https://royalsocietypublishing.org/doi/full/10.1098/rspb.2021.1918>
<https://www.pnas.org/doi/abs/10.1073/pnas.1404167111>
<https://journals.plos.org/plosgenetics/article?id=10.1371/journal.pgen.1008145>
<https://www.journals.uchicago.edu/doi/full/10.1086/716014>
<https://paleoanthropology.org/ojs/index.php/paleo/article/view/114>
<https://www.sciencedirect.com/science/article/pii/S1040618218311698>

- We have incorporated a discussion of these references in our paper in the first paragraph of Section VI-A, the second half of the first paragraph of Section VII, and the last paragraph of Section VIII.

Reviewer #2 (Remarks to the Author):

The authors present a logical hypothesis regarding an additional source of increased energy availability required to facilitate brain expansion in the human lineage – fermented foods. Their descriptions are thorough, and I appreciate their suggestions for future comparative studies. However, this manuscript is missing a discussion of work that has already focused on the importance of fermented foods in nonhuman ape and human evolution, including: 1) alterations to the ADH4 gene that improve ethanol digestion (from fermented fruits) in apes (Carrigan et al. 2015, PNAS); 2) coevolution between fermented foods and sour taste abilities (Frank et al. 2022, Prob B); and 3) the possibility

that Neanderthals fermented meat to maintain vitamin C content and avoid scurvy (Speth 2019, Quarterly Int'1).

- Thank you for taking the time to review our article and for offering these suggestions. We have incorporated coverage of these papers into the article: Section VI-A, second half of paragraph 1; Section V, second half of paragraph 1; and Section VIII, middle of paragraph 2). We also added a paragraph (last paragraph of section VIII) that discusses evidence for genetic changes in human and ape lineages that may be related to adaptations for the consumption of fermented foods.

REVIEWERS' COMMENTS:

Reviewer #1 (Remarks to the Author):

Thank you for the careful revisions of the manuscript. I believe it is greatly improved. One final point is that some of the papers that are now cited raise the possibility of the 'external fermentation hypothesis'. In that sense, it seems somewhat incorrect to label that hypothesis as novel in this paper. I suggest instead labeling it as a 'recent' hypothesis and say that you are exploring it in more depth in this paper.

RESPONSE

Reviewer #1 (Remarks to the Author):

Thank you for the careful revisions of the manuscript. I believe it is greatly improved. One final point is that some of the papers that are now cited raise the possibility of the 'external fermentation hypothesis'. In that sense, it seems somewhat incorrect to label that hypothesis as novel in this paper. I suggest instead labeling it as a 'recent' hypothesis and say that you are exploring it in more depth in this paper.

Response to Reviewer #1:

We thank the Reviewer for their careful reading of our most recent version. We agree that recent papers that have come out in the last few years have also explored the possibility of a similar mechanism accounting for dietary change in human evolution. As the reviewer suggests, we have removed language in the manuscript that refers to our hypothesis as “new”, “novel”, or “recent” and simply describe in detail our perspective on this framework